# Psychological Health in Late Effects of Poliomyelitis: Ten-Year Follow-Up

**DOI:** 10.3390/healthcare11243144

**Published:** 2023-12-11

**Authors:** Shimon Shiri, Anat Marmor, Morad Jalagil, Hagai Levine, Isabella Schwartz, Zeev Meiner

**Affiliations:** 1Faculty of Medicine, The Hebrew University, Jerusalem 91120, Israel; anatmar@hadassah.org.il (A.M.); dr_mjalajel@live.com (M.J.); isabellas@hadassah.org.il (I.S.); meiner@hadassah.org.il (Z.M.); 2The Departments of Physical Medicine & Rehabilitation, Hadassah Mount Scopus Hospital, Jerusalem 91120, Israel; 3Hadassah Braun School of Public Health, The Hebrew University, Jerusalem 91120, Israel; hlevine@hadassah.org.il

**Keywords:** late effect of poliomyelitis, psychological health, hope, life satisfaction, work, subjective health perception

## Abstract

Background: Individuals with late effects of poliomyelitis (LEoP) cope with various physical and psychological symptoms throughout their entire life which become more severe as they are ageing. Objectives: To perform a 10-year follow-up of the functional status and levels of psychological health of individuals with LEoP and to examine the associations of hope levels, work status, health perceptions, and life satisfaction with functional and psychological changes. Design: A within-subject 10-year follow-up study. Participants: Eighty-two individuals with LEoP who participated in a previous study 10 years ago. Methods: Outcome measures included the functional status of individuals with LEoP assessed by the activities of daily living (ADL) questionnaire, emotional distress based on the Global Health Questionnaire (GHQ), hope based on the Hope Scale, life satisfaction as measured by the Satisfaction with Life Scale (SWLS), and subjective health perception. The McNemar test, paired *t*-test, Spearman’s correlation coefficient, and linear regression were used for statistical analysis. Results: The mean age was 66.9 ± 8.5 years with a male–female ratio of 0.52. A significant functional deterioration was noticed during the follow-up years. Yet, the functional deterioration was not associated with changes in psychological health. Psychological health was associated with elevated levels of hope and life satisfaction. Individuals with LEoP who continued to work demonstrated higher psychological health, higher levels of hope, and greater life satisfaction. Conclusions: Individuals with LEoP demonstrated significant psychological health, manifested in their ability to block emotional distress and maintain life satisfaction despite the deterioration in their functional status. Hope and psychological health were associated with increased life satisfaction. Work appeared to be a significant source of psychological health in this population.

## 1. Introduction

The World Health Organization estimated that 12–20 million people worldwide are living with sequelae of poliomyelitis [1]. For decades, individuals with late effects of poliomyelitis (LEoP) have been affected by significant long-term disability compared to the general population [2,3,4], especially those who were diagnosed with post-polio syndrome [5]. In Israel, there are more than 2500 individuals with LEoP, most of them in the seventh decade of their life, and lately their condition has been deteriorating because of accelerated aging processes and post-polio syndrome (PPS) [6].

Long-term disability is often associated with a significant chronic decline in life satisfaction and with increased distress levels [7]. In individuals with different types of disabilities, greater levels of disability have been associated with decreased perceived quality of life, particularly in individuals using wheelchairs [8].

Yet, the situation of individuals with LEoP seems to be different and more encouraging. In one study [9], individuals with LEoP were found to show levels of sense of coherence (SOC) similar to those of individuals without disabilities. SOC [10] reflects an adaptive orientation that enables one to comprehend, manage, and attach meaning to stressogenic events and therefore successfully cope with adverse situations such as disease. In a later study [11], SOC was correlated with better perceptions of self and greater life satisfaction. Further, it seems that individuals with LEoP apply problem-focused and emotion-focused coping, which adapts when dealing with different stressors [12]. When looking at other cultures, as reflected in a qualitative study among Polio survivors in northwest Nigeria, it may be noted that spirituality and self-perception are significant in determining their quality of life [13].

Similarly, although their life satisfaction was lower than that of the general population [14,15], individuals with LEoP presented optimistic views with regard to their life [16]. In a recent study, participants reported how they developed optimistic strategies, such as building a robust self-image, learning to trust their ability to manage challenging situations, and focusing on positive aspects of their life, such as their achievements [16]. In an earlier study, Swedish individuals with LEoP reported overall satisfaction with their life and with specific domains such as vocation, partner relationship, and family life [16,17,18].

In a previous study [18], we reported that individuals with LEoP presented decreased a psychological health and lower quality of life than the controls. Participants tended to use optimistic strategies, particularly hope, which had generally been associated with adaptive rehabilitation processes [19] and improved quality of life. In a follow-up study, we found that individuals with LEoP demonstrated increased psychological health despite the deterioration in their physical health [20]. In both studies, work appeared to be a significant source of psychological health and hope [18,20].

Our previous studies were conducted ten years ago. Since then, the poliomyelitis population has aged and experienced a decline in clinical and functional levels. It is not clear, however, whether this deterioration affected also their psychological health or whether they maintained their unique resilience, similarly to what was found in previous studies. Therefore, in the present study, we conducted a 10-year follow-up of the association between the functional status and psychological health of individuals with LEoP. To explore what mechanisms contribute to the psychological endurance of the LEoP population, we examined the associations between levels of hope, work status, health perceptions, and life satisfaction. These variables were chosen because of their association with health behaviors and endorsement. In particular, hope and life satisfaction have been found to be associated with healthier behaviors and disease treatment among individuals coping with chronic disease [21,22].

## 2. Methods

### 2.1. Participants

In 2009, we conducted a study of 197 patients registered in the database of the post-polio clinic at Hadassah Mount Scopus Medical Center in Jerusalem [23]. For the current study, a text message was sent to all these patients, and those who answered were scheduled for a phone interview to complete the survey (the duration of the interview was about 45 min). Inclusion criteria were a diagnosis of previous paralytic poliomyelitis, with confirmation by clinical and laboratory examination, and validation of normal mental functioning, as shown by a score of 8 or above in the short portable mental status questionnaire (SPMSQ) [24]. Exclusion criterion was evidence of other neurological diseases that could explain the neurological impairment. All participants provided consent by agreeing to complete the questionnaires.

All the questionnaires were translated into Hebrew and validated. For Arabic-speaking participants, questionnaires were translated into Arabic by a native speaker, and then translated back into the source language of the questionnaires. In the final stage, the translated and original questionnaires were compared to identify discrepancies and to reach an agreed format. The study was approved by the Institutional Review Board.

### 2.2. Survey Instruments

Demographic questionnaire. We obtained demographic data, including age, gender, family, employment, and physical activity status.

Mental functioning. The short portable mental status questionnaire (SPMSQ) is a 10-item questionnaire that has been tested, standardized, and validated with a population of older adults [24]. Eight or more correct answers are considered to represent normal mental functioning. The Cronbach’s alpha coefficient of the SPMSQ is 0.88 [25,26].

Functional status. We used a self-report questionnaire to identify difficulties in performing activities of daily living (ADL). The ADL questionnaire assesses the level of independence in bathing/showering, dressing, eating, and functional mobility, which includes mobility in bed, sit-to-stand transfers, and mobility in and out of doors. The patients’ overall functioning was assigned a grade on a five-point scale: 1 = complete independence, 2 = independent with difficulties, 3 = independent with assistance, 4 = requiring personal assistance, and 5 = fully dependent. We also inquired about walking aids and the use of wheelchair.

Psychological health. The General Health Questionnaire (GHQ-12) is a 12-item questionnaire assessing current mental health by measuring emotional distress (ED) [27]. Items of this instrument serve as measures of satisfaction and sense of general competence. Six of the items are phrased positively and six negatively. Each item is rated on a four-point scale: 1 = never, 2 = rarely, 3 = sometimes, and 4 = very often. To normalize the positively and negatively phrased items, the scale order has been adjusted and the scores were calculated as 49 minus the final score. Therefore, the scores ranged between 1 (49–48) and 37 (49–12), with higher scores indicating increased distress. The Cronbach’s alpha of the GHQ-12 is 0.9 [28,29]

Hope. The Hope Scale [30] is a valid, internally and temporally reliable trait self-report measure that includes 12 items, with participants rating their response on an eight-point Likert scale. Conceptually, hope reflects a cognitive set of appraisals that individuals perform with regard to goal-related activities. Two dimensions exist within this cognitive domain: agency and pathways. Agency relates to the motivational components necessary for the consistent and sustained efforts needed to achieve goals. Pathways refer to the perceived methods for achieving established goals. The Hope Scale has been used in a variety of studies, including health-related environments [31]. The score for each respondent was based on the average of their responses on the 12 items. The Cronbach’s α of the Hope Scale is 0.93 [30].

Life satisfaction. The Satisfaction with Life Scale (SWLS) was developed by Diener and colleagues; it was translated into Hebrew and validated [32,33]. This valid self-report measure includes five items, with participants rating their responses on a seven-point Likert scale. The score for each respondent was the average of their responses on the five items, a higher score indicating greater satisfaction. The Cronbach’s α of the Satisfaction with Life Scale (SWLS) is 0.87 [34].

Subjective health perception. Patients were asked to rate their physical, emotional, and overall health state on a five-point Likert scale: 1 = excellent, 2 = very good, 3 = good, 4 = fair, and 5 = poor. The score for each respondent was the average of their responses on the 3 items, a higher score indicating lower health perception. This questionnaire is based on the World Mental Health Survey Initiative, using the Composite International Diagnostic Interview (CIDI).

### 2.3. Statistical Methods

Descriptive statistics for continuous variables are presented as mean and standard deviation (SD), and for the categorical variables as frequency and percent. We used the McNemar test to compare categorical data of 2009 and 2020 responses. We calculated and presented mean and standard deviation (SD) scores of functional, general health, and subjective health questionnaires in 2009 and 2020. We used a paired *t*-test to compare these time periods. We also calculated mean and standard deviation (SD) and alpha Cronbach’s scores of the Hope Scale and life satisfaction at the 2020 follow-up. We calculated Spearman’s correlations to determine the correlation between the different questionnaires and health status. We performed a linear regression for the association between SWLS and working status, as well as for other questionnaires, including ADL, IPPS, general and subjective health, and the Hope Scale. Statistical significance was set at *p* < 0.05.

## 3. Results

### 3.1. Demographics

Of 197 patients who met the study criteria and participated in the previous study, 82 confirmed their participation in the current study, 9 were deceased, and 106 were not traced for various reasons. The demographic data of the 82 participants in both studies (2009 and 2020) are shown in Table 1. The mean age was 67 years, 52.4% were men, 79.3% were Jews, and 30.5% were still employed. A statistically significant difference was found in the percentage of participants with LEoP who were employed, which dropped from 43.9% in 2009 to 30.5% in 2020. When we compared the demographic data of 82 patients participated in the two studies with 115 that were lost to follow up, no statistical differences were found).

### 3.2. Comparison of ADL Functions, GHQ, and Subjective Health Perception between the Current Study and the Previous One

The results of a comparison of average functioning in ADL, psychological health as measured by GHQ, and health perception are shown in Table 2. The average ADL score increased significantly in the 10-year follow-up (from 1.97 ± 0.64 to 2.22 ± 0.85, *p* < 0.001), reflecting the functional deterioration of people with LEoP in the course of the decade. At the same time, the psychological health of people with LEoP, as assessed by the GHQ, remained unchanged, similarly to the participants’ subjective perception of health. Hope and SWLS were not measured in the previous study. In our study, the Cronbach’s α of the GHQ-12 is 0.86, of the Hope Scale is 0.61, and of the SWLS is 0.76.

### 3.3. Correlation between ADL Functions, GHQ, Subjective Health Perception, and the Hope Scale and SWLS

To elucidate what mechanisms contribute to the dissociation between deterioration in functional level and the endurance in the psychological status, we used Spearman’s correlation coefficient test to further examine the associations between levels of hope, life satisfaction, ADL, GHQ, and health perceptions. The results are shown in Table 3. Individuals with LEoP whose ADL functioning was low showed decreased psychological health, reflected in higher GHQ scores and lower health perception. No significant correlation was found between ADL functioning, life satisfaction, and levels of hope. As expected, lower psychological health correlated with less hope, lower life satisfaction, and lower subjective health perception. Levels of hope were positively associated with life satisfaction and better subjective health perception.

### 3.4. Linear Regression Analysis

To determine the effect of specific variables on life satisfaction, we conducted regression analysis. Because of the high correlation between hope and GHQ, we analyzed them separately before combining them into a full model. As shown in Table 4, hope and GHQ both showed a significant association with life satisfaction in the partial models. In the full model, hope and higher health perception had a positive significant effect on life satisfaction, whereas GHQ had no significant effect on life satisfaction beyond the effect of hope. ADL and occupation status had no significant effect.

### 3.5. Comparison between Individuals with LEoP Who Work and Those Who Do Not

We conducted a comparison to reveal the differences in psychological variables between individuals with LEoP who worked at the time of the study and those who did not. As shown in Table 5, psychological health, levels of hope, life satisfaction, and health perceptions were higher in workers than in non-workers.

## 4. Discussion

The main objectives of the present study were to perform a 10-year follow-up of the functional status and levels of psychological health of individuals with LEoP and to examine the associations between hope levels, work status, health perceptions, and life satisfaction with functional and psychological changes. The results of this study indicate that individuals with LEoP managed to retain their psychological health despite the decline in their functional status. This finding is a significant indication of their psychological resilience and is consistent with previous studies showing that individuals with LEoP are satisfied and hold optimistic views about their life and achievements [16,17]. The current findings are consistent with those of our previous study [20], showing that participants managed to retain their psychological health despite deteriorating physical health.

The prospective, within-subject design of the present study, ten years after the first study, suggests that the positive views held by individuals with LEoP are enduring and constant. Such attitudes may be unique to individuals with LEoP, given that many other individuals with long-term disabilities experience decline not only in their physical but also in their psychological state and general wellbeing [7,8]. Similar results to those of the current study were found in a large Norwegian study, in which individuals with LEoP reported only minor subjective decline in health and wellbeing compared with their reports 20 years earlier, despite deterioration in their physical state [35].

The results also indicate that the subjective health perception of the participants has not changed in the ten years that elapsed between the two studies, despite their physical decline. Participants appear to have ascribed little weight to the late effects of their condition. This provides a hint about their coping strategies: in this case, a tendency to minimize difficulties and emphasize the positive aspects of their condition. This strategy has been noted in an earlier study, in which individuals with LEoP tended to ignore negative facets of their life and disability, focusing on positive aspects, such as their achievements [16]. Another important finding of the present study is that the reduced functional status of individuals with LEoP was not associated with decreased life satisfaction.

The associations between levels of hope, work status, health perceptions, and life satisfaction were also addressed. Hope was found to be positively associated with higher psychological health and greater levels of life satisfaction. With regard to work, it seems that working serves as a source of strength for individuals with LEoP. Although only 30.5% of participants were still working, this rate was high when taking into consideration their advanced age and disabilities. We found that individuals with LEoP who have still worked reported higher levels of psychological health, hope, life satisfaction, and health perception than did those who did not work. Previous studies of patients with PPS and other chronic diseases have shown a positive association between work and quality of life [36,37]. In a previous study [18], we found a high association between employment and improved physical and mental quality of life in individuals with PPS. Work has been an important long-term goal of individuals with LEoP [38], although they have never reached the levels of employment of the general population [23,39].

When trying to explore the underlying factors that may explain these associations, the concept of “career adaptability” emerges. Career adaptability refers to a set of individual resources to cope with developmental tasks, to participate in working life, and to adapt to changes in work and job conditions [40]. It may be seen as the process by which people actively construct their career life, while coping with changing situations that they experience in their social contexts [38,41]. Career adaptability is composed of four resources and coping strategies. First: concern for the future, Second: control-holding a belief that the future is in part manageable, Third: curiosity or exploring the environment and to acquire information on oneself and the outside world, and Fourth: self-confidence, which is the self-efficacy and own ability to cope with challenges and obstacles that may be inhibit one’s goals. These attributes resemble many of the characteristics that are attached to the struggle of individuals with LEoP [42].

Taking into account the importance of employment for the psychological health and life satisfaction of individuals with LEoP found in this and previous studies, it is of utmost importance to develop customized prevention and rehabilitation programs to maintain the level of function and ability to work in this population. An active program that enables people with LEoP to maintain their working position and to enhance their hope level may be beneficial in order to improve their psychological health and quality of life.

This study has several limitations. The number of patients who participated in both the original and the present study was relatively small, reflecting the difficulties in reaching all patients and significant attrition between the two studies. The fact that only 44% of the patients who were still alive at the time of the present study participated could possibly produce a selection bias and decrease the external validity of the study. Yet, given the patients’ advanced age and the 10 years that have elapsed since the initial study, this rate of participation was expected. The fact that hope and life satisfaction were not tested in 2009 is a limitation, and the causal effect between hope, psychological health, and life satisfaction remains to be tested. Furthermore, poliomyelitis patients who attended our post-polio clinic may have presented with more severe symptoms than the general population of Israeli poliomyelitis patients. This may have produced another selection bias, which may also reduce the external validity of the findings. Additionally, because our survey was conducted by phone, the risk of confirmation bias is increased. In general, since the results are based on a sample of patients who attended our clinic who may have local cultural characteristics, one should be cautious when generalizing the results to other populations or cultures. Future studies with individuals with LEoP from different geographical areas and cultures may assist in finding out to what degree the present findings can be generalized.

## 5. Conclusions

In conclusion, several studies have highlighted the optimistic and resilient nature of individuals with LEoP. The findings of the present study further corroborate the findings of these earlier studies. The current results indicate that individuals with LEoP manage to maintain their psychological health in the long term and towards their seniority, despite a decline in their functional status. The findings also indicate that hope has been distinctively associated with better psychological health among these individuals with LEoP.

Finally, the findings also suggest that individuals with LEoP who have been still working reported better psychological health and subjective life satisfaction than those who have retired, demonstrating the significance of maintaining employment for this population. It is suggested to perform similar studies in patients suffering from chronic disabilities due to other neurological diseases and to compare with the results of people LEoP, as well as to repeat the study in people with LEoP of more advanced ages.

## Figures and Tables

**Table 1 healthcare-11-03144-t001:** Demographic characteristics of the 82 participants in both the 2009 and 2020 studies.

	Present Study 2020 (n = 82)	Previous Study 2009 (n = 82)	*p* Value
Age (years)	66.9 ± 8.49	57.2 ± 8.8	
Male/Female	43 (52.4%)/39 (47.6%)	43 (52.4%)/39 (47.6%)	
Jews%/Arabs%	65 (79.3%)/17 (20.7%)	65 (79.3%)/17 (20.7%)	
Married%	65.9%	73.4%	0.21
Mean years of education	13.5 ± 4.3	13.5 ± 4.3	
Working in the last week%	30.5%	43.9%	0.03

**Table 2 healthcare-11-03144-t002:** Comparison of average ADL, GHQ, Hope, SWLS, and health perception between the 2009 and 2020 studies.

	Present Study 2020 n = 82 (Mean ± SD)	Previous Study 2009 n = 82 (Mean ± SD)	*p* Value
ADL	2.22 ± 0.85	1.97 ± 0.64	<0.001
Sum GHQ	10.26 ± 7.08	10.12 ± 6.11	0.7701
Health perception	3.26 ± 1.00	3.15 ± 0.70	0.260
Hope	2.19 ± 0.40	ND	
SWLS	4.72 ± 1.22	ND	

ADL = Activity of daily living; GHQ = Global health questionnaire; SWLS = Satisfaction with Life Scale. ND = Not done.

**Table 3 healthcare-11-03144-t003:** Inter-correlations between ADL, GHQ, Hope, SWLS, health perception in 82 patients with LEoP in the present study (Spearman’s correlation coefficient test).

	ADL	GHQ	Hope	SWLS	Health Perception
ADL	1.0				
GHQ	0.341 **	1.0			
Hope	−0.195	−0.639 **	1.0		
SWLS	−0.196	−0.479 **	0.560 **	1.0	
Health perception	0.274 *	0.674 **	−0.597 **	−0.540 **	1.0

ADL = Activity of daily living; GHQ = Global health questionnaire; SWLS = Satisfaction with Life Scale. * Correlation is significant at the 0.05 level (2-tailed). ** Correlation is significant at the 0.01 level (2-tailed).

**Table 4 healthcare-11-03144-t004:** Linear regression with SWLS as a dependent variable and current ADL, GHQ, hope, and health perception as predictor variables.

	Without Hope		Without GHQ		Full Model	
Predictor	Βeta (95% CI)	*p* Value	Βeta (95% CI)	*p* Value	Βeta (95% CI)	*p* Value
Average ADL	−0.08 (−0.35–0.19)	0.5723	−0.08 (−0.33–0.17)	0.5227	−0.08 (0.18–0.35)	0.5305
Average hope			1.08 (0.44–1.71)	0.0011	0.89 (0.68–0.11)	0.0258
Total GHQ	−0.06 (−0.11–0.01)	0.0114			−0.03 (−0.08–0.03)	0.3124
Health perception	−0.42 (−0.72–0.13)	0.0056	−0.36 (−0.62–0.11)	0.0063	−0.34 (−0.64–0.04)	0.026
Work	0.42 (−0.04–0.88)	0.0728	0.29 (−0.15–0.74)	0.1942	0.31 (−0.15–0.76)	0.184

ADL = Activity of daily living; GHQ = Global health questionnaire; SWLS = Satisfaction with Life Scale.

**Table 5 healthcare-11-03144-t005:** Difference between individuals with LEoP who worked and those who did not.

	Average Hope	Average SWLS	Total GHQ	Average ADL	Average Health Perception
Workers (N = 36) mean (SD)	2.36 (0.33)	5.22 (1.11)	7.91 (5.56)	2.12 (0.84)	2.87 (1.03)
Non-workers (N = 46) mean (SD)	2.07 (0.41)	4.38 (1.14)	12.05 (7.63)	2.29 (0.86)	3.52 (0.87)
Estimated difference	−0.28 (0.38)	−0.84 (1.13)	4.14 (6.82)	0.17 (0.85)	0.65 (0.94)
Upper	0.4535	1.3476	−1.093	0.212	1.0688
Lower	0.1122	0.3365	−7.182	−0.5415	0.2339
Sig	0.002	0.001	0.01	0.38	0.003

ADL = Activity of daily living; GHQ = Global health questionnaire; SWLS = Satisfaction with Life Scale.

## Data Availability

Data are contained within the article.

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
