# Peer review of "Psychological Health in Late Effects of Poliomyelitis: Ten-Year Follow-Up"

_healthcare, 2023, doi:10.3390/healthcare11243144_

Round 1

Reviewer 1 Report

Comments and Suggestions for Authors

Exploration of Underlying Causes: It is crucial for the authors to consider the possibility of underlying causes for the identified associations in the study. While relationships between hope, employment status, and life satisfaction have been observed, it is important to discuss what drives these associations. The authors could point out the potential existence of mechanisms or intermediary factors that explain these relationships. This analysis would provide more detailed information for future interventions or studies.

Emphasis on Generalization Limitation: Emphasizing the fact that the study's results are based on patients who attended a clinic in Israel is essential. As different regions and cultures may exhibit significant variations in terms of beliefs, practices, and support systems, it is fundamental to acknowledge the limitation of generalizing the results to other populations. The authors can highlight this limitation in the results and discussion sections, underscoring the need for future studies that consider a variety of cultural and geographical contexts to assess the applicability of the findings in different scenarios. 

Conclusion Section: The conclusion of the study should be organized into a dedicated section for clarity and ease of reference.

Author Response

Reviewer 1 – Thank you, Reviewer 1, for your valuable remarks. 

Exploration of Underlying Causes: It is crucial for the authors to consider the possibility of underlying causes for the identified associations in the study. While relationships between hope, employment status, and life satisfaction have been observed, it is important to discuss what drives these associations. The authors could point out the potential existence of mechanisms or intermediary factors that explain these relationships. This analysis would provide more detailed information for future interventions or studies.

We have added a paragraph in response to the the reviewer's remark, that may add light to relationships between hope, employment status, and life satisfaction have been observed,  Lines 273-299.

Emphasis on Generalization Limitation: Emphasizing the fact that the study's results are based on patients who attended a clinic in Israel is essential. As different regions and cultures may exhibit significant variations in terms of beliefs, practices, and support systems, it is fundamental to acknowledge the limitation of generalizing the results to other populations. The authors can highlight this limitation in the results and discussion sections, underscoring the need for future studies that consider a variety of cultural and geographical contexts to assess the applicability of the findings in different scenarios. 

We have emphasized in the limitations section a remark concerning the limitations that exist due to the locality and cultural homogeneity. Lines 318-323.

 Conclusion Section: The conclusion of the study should be organized into a dedicated section for clarity and ease of reference.

The conclusion section has been re-written following this reviewer's remark. Lines 324-336.

Reviewer 2 Report

Comments and Suggestions for Authors

I would like to congratulate the authors for the manuscript they present, but in my opinion it can be improved. My comments are organized below for your consideration. I hope my comments are useful for the study author(s) and editorial staff.

#1 Introduction:

This section seems adequate for the objectives of the study, but is not contextualised with the present theoretical background (about 79% the articles cited are over 5 years old), so i think the theoretical background could be improved by citing more articles less than 5 years old.

The citations (4; 7; 17; 19) are all from articles published by the authors, although I consider your knowledge in this area to be excellent, in order to enrich the literature of your manuscript and support diversity, I advise you to cite more articles from other renowned scholars in this field.

#4 Discussion

The discussion is very descriptive, not very analytical and not very consistent, in general, is not supported by recent studies, as the vast majority (about 95%) of the studies referred to are over 5 years old, so I think it would be important to cite more recent studies, less than 5 years old and the discussion should be more reflective and analytical.

Line: 216 – 217: “The current findings are consistent with those of our previ-216 ous study [19]” - it would be important to refer to studies by other authors.

# Conclusions

There should be a chapter for conclusions and the conclusions presented by the authors, in my opinion, do not show their added value.

# References

- The authors should update the references, as around 89% are more than 5 years old.

Author Response

Reviewer 2 –. Thank you, reviewer, 2 for your valuable remarks.

#1 Introduction:

This section seems adequate for the objectives of the study, but is not contextualised with the present theoretical background (about 79% the articles cited are over 5 years old), so i think the theoretical background could be improved by citing more articles less than 5 years old.

Recent papers from different years were added to the references list. 

The citations (4; 7; 17; 19) are all from articles published by the authors, although I consider your knowledge in this area to be excellent, in order to enrich the literature of your manuscript and support diversity, I advise you to cite more articles from other renowned scholars in this field.

We have added two significant studies from other groups to the list of citations, e.g:

Nolvi, M., Brogårdh, C., Jacobsson, L., & Lexell, J. (2022). Sense of coherence and coping behaviours in persons with late effects of polio. Annals of Physical and Rehabilitation Medicine, 65(3), 101577.‏

Sulaiman, S. K., Aldersey, H. M., Fayed, N., & Kaka, B. (2021). Exploring the perception of quality of life of polio survivors in Northwest Nigeria. Applied Research in Quality of Life, 16, 1369-1389.‏

#4 Discussion

The discussion is very descriptive, not very analytical and not very consistent, in general, is not supported by recent studies, as the vast majority (about 95%) of the studies referred to are over 5 years old, so I think it would be important to cite more recent studies, less than 5 years old and the discussion should be more reflective and analytical.

Line: 216 – 217: “The current findings are consistent with those of our previous study [19]” - it would be important to refer to studies by other authors.

We added 5 recent studies from different groups

# Conclusions

There should be a chapter for conclusions and the conclusions presented by the authors, in my opinion, do not show their added value.

The discussion section was reorganized in the light of the reviewer's remarks

# References

- The authors should update the references, as around 89% are more than 5 years old.

5 recent papers from different groups were added to the references list. 

Reviewer 3 Report

Comments and Suggestions for Authors

First of all, congratulations on the topic covered which is so sensitive, but also so necessary.

Overall well written but some sections need  clarification:

In the introduction part, argue why you included the Hope and SWLS variables in study 2, what contribution they make to study 1 and the literature.

In the method part, specify how much time was needed to complete the instruments.

In the Survey instruments section provide information on the internal consistency of instruments reported in other studies. And in the results section provide information on the internal consistency of the scales in the present study.

In the Statistical methods section present information about the method for testing the internal consistency, as well as testing the normal distribution of the variables, and in the Results section briefly present the results.

Table 1 could be made in a more reader-friendly form, specifically the data on the Male/Female and Jews %/Arabs % rows, showing the data separately for both studies.

In the Discussion part begin by presenting the main objective of the study and then continue with the results. Also in this section you describe the implications your results have for practice, how they can be used to improve the lives of people with LEoP.  What are the future research directions based on the results of the study and its limitations?

Congratulations on an ambitious project and significant contributions to the field.

Author Response

Reviewer 3 – Thank you, reviewer 3 for your valuable remarks.

 Overall well written but some sections need clarification:

In the introduction part, argue why you included the Hope and SWLS variables in study 2, what contribution they make to study 1 and the literature.

We have added in the introduction an explanation as well as references in which the significance of Hope and SWLS is addressed. Lines 86-89

In the method part, specify how much time was needed to complete the instruments.

The duration of the interview was about 45 minutes (line 96)

In the Survey instruments section provide information on the internal consistency of instruments reported in other studies. And in the results section provide information on the internal consistency of the scales in the present study.

The information regarding the internal consistency of instruments was added to the Survey instruments section [lines 114- 148] and the internal consistency of the scales in the present study was added to the results section. 

In the Statistical methods section present information about the method for testing the internal consistency, as well as testing the normal distribution of the variables, and in the Results section briefly present the results.

We added in line 163 that we preformed alpha cronbach section. We did perform a normal distribution analysis and most of the variables were normally distributed.

Table 1 could be made in a more reader-friendly form, specifically the data on the Male/Female and Jews %/Arabs % rows, showing the data separately for both studies.

Table 1 was changed according to the reviewer suggestion

In the Discussion part begin by presenting the main objective of the study and then continue with the results. Also in this section you describe the implications your results have for practice, how they can be used to improve the lives of people with LEoP.  What are the future research directions based on the results of the study and its limitations?

The beginning of the discussion was changed according to the reviewer suggestion. An implication to improve the quality of life of LEoP was added in lines 304-306. We suggested some future research direction on lines 335-337.